# Anti-Inflammatory and Antioxidant Capacity of a Fruit and Vegetable-Based Nutraceutical Measured by Urinary Oxylipin Concentration in a Healthy Population: A Randomized, Double-Blind, Placebo-Controlled Clinical Trial

**DOI:** 10.3390/antiox11071342

**Published:** 2022-07-08

**Authors:** Raúl Arcusa, Juan Ángel Carrillo, Begoña Cerdá, Thierry Durand, Ángel Gil-Izquierdo, Sonia Medina, Jean-Marie Galano, Débora Villaño Valencia, Javier Marhuenda, Pilar Zafrilla

**Affiliations:** 1Faculty of Health Sciences, Universidad Católica de San Antonio, 30107 Murcia, Spain; rarcusa@ucam.edu (R.A.); jacarrillo4@alu.ucam.edu (J.Á.C.); bcerda@ucam.edu (B.C.); dvillano@ucam.edu (D.V.V.); mpzafrilla@ucam.edu (P.Z.); 2Institut des Biomolécules Max Mousseron (IBMM), Pôle Chimie Balard Recherche, UMR 5247, National School of Chemistry Montpellier, Université de Montpellier, Centre National de la Recherche Scientifique, 1919 Route de Mende, CEDEX 05, 34293 Montpellier, France; thierry.durand@umontpellier.fr (T.D.); jgalano@univ-montp1.fr (J.-M.G.); 3Research Group on Quality Safety and Bioactivity of Plant Foods, Food Science and Technology Department, CEBAS-CSIC, 30100 Murcia, Spain; angelgil@cebas.csic.es (Á.G.-I.); soniamedes@gmail.com (S.M.)

**Keywords:** oxidative stress, metabolic syndrome, inflammation, metabolic diseases, isoprostanes, oxylipins

## Abstract

Oxylipins, lipid biomarkers of inflammation are considered the gold standard method to evaluate the inflammatory and antioxidant status. The aim of the present study was to investigate whether the administration of a polyphenolic extract shot in the form of a nutraceutical was able to reduce inflammation, measured in urine markers. Ninety-two participants (45 males, 47 females, age 34 ± 11 years, weight 73.10 ± 14.29 kg, height 1.72 ± 9 cm, BMI 24.40 ± 3.43 kg/m^2^) completed the study after an intervention of two 16-week periods consuming extract or placebo separated by a 4-week washout period. The results showed significant differences in terms of reduction of different pro-inflammatory oxylipins (15-keto-PGF_2α_ (from 0.90 ± 0.25 ng/mL to 0.74 ± 0.19 ng/mL *p* < 0.05), *ent*-PGF_2α_ (from 1.59 ± 0.37 ng/mL to 1.44 ± 0.32 ng/mL *p* < 0.05), 2,3-dinor-15-F_2t_-Isop) (from 1.17 ± 0.35 ng/mL to 1.02 ± 0.27 ng/mL *p* < 0.05), in total oxylipins count (from 8.03 ± 1.86 ng/mL to 7.25 ± 1.23 ng/mL *p* < 0.05), and increase in PGE_2_ (from 1.02 ± 0.38 ng/mL to 1.26 ± 0.38 ng/mL *p* < 0.05) which has an anti-inflammatory character, after extract consumption compared to placebo. The available data seem to indicate that long-term consumption of a nutraceutical with high polyphenol content improves inflammation and oxidation parameters measured in urine, through UHPLC-QqQ-ESI-MS/MS.

## 1. Introduction

Oxylipins are bioactive oxygenated lipids [1], characterized as a group of metabolites derived from the oxidation of polyunsaturated fatty acids (PUFA) with multiple biological functions that play an important role in inflammation, immunity, vascular functions, etc. [1,2]. Within the oxylipins, the eicosanoid family includes isoprostanes (IsoPs), prostaglandins (PG), leukotrienes (LK) and thromboxanes (TX), all of which are lipid mediators involved in the pathophysiology of cells, organs and tissues [3,4,5]. Currently, more than 100 types of oxylipins with various overlapping and interconnected functions have been identified [2]. To summarize, prostanoids include PG and TX which are metabolized through the cyclooxygenase (COX) enzymatic pathway, LK through the lipoxygenase (LOX) enzymatic pathway [6] and IsoPs by free radical action through a COX-independent non-enzymatic pathway [7,8], comprehensively described and illustrated in the publication by Libia et al. [4]. For the prostanoids, through COX, Arachidonic Acid (AA) is metabolized to PGG_2_, which is quickly metabolized to PGH_2_, PG precursor of four primary bioactive PGs (PGD_2_, PGE_2_, PGI_2_, and PGF_2α_) that give name to the D, E, I and F pathways, respectively, and are metabolized to numerous bioactive metabolites [9]. Oxylipins are produced after damage or stimulus and given their short half-life, they are not stored, but are synthesized de novo in a strictly regulated manner [5]. After activation by such a stimulus, PUFA at the sn-2 position of glycerophospholipids in cell membranes are released by the action of phospholipase A_2_ (PLA2) [10], being oxygenated by one of the different families of COX, LOX and cytochrome P450 (CYP) [11]. The type of oxylipin synthesized will depend largely on the PUFA (omega 3 or omega 6) that are oxidized and present in the membrane, as well as the oxygenase that oxidizes it [2]. Elevated values of oxylipins have been described in different cardiovascular pathologies related to inflammation such as diabetes, thrombosis, hypertension and hyperlipidemia [12]. Once synthesized, oxylipins can exert paracrine or autocrine action. They can diffuse across the plasma membrane, signal/bind to G protein-coupled receptors (GPCRs) or activate peroxisome proliferator-activated receptors (PPARs) or ligand-activated transcription factors [13,14].

The scientific literature has widely documented the role played by polyphenols in health, attributing to them various properties such as antioxidant, immunomodulatory, anticancer, anti-inflammatory and cardiovascular protection [15,16,17,18]. Such polyphenols are bioactive compounds synthesized as secondary metabolites of plants, acting as a defence mechanism against adverse conditions/aggression [19]. Bioactive compounds present in fruits and vegetables seem to be responsible for giving them pharmacological activities as well as characteristic flavour and colour [18]. Polyphenols are categorized according to their chemical structures into flavonoids (chalcones, anthocyanidins, proanthocyanins, flavonols, neoflavonoids, flavones and isoflavones) and non-flavonoids (stilbenes, phenolic amides and phenolic acids) [20,21,22], with more than 8000 structural variants. Polyphenols have antioxidant capacity, they have the ability to inhibit enzymes involved in the production of eicosanoids (COX and LOX), as well as Xanthine oxidase, NADPH oxidase (NOX), enzymes involved in the production of reactive oxygen species (ROS) known as free radicals, while modulating endogenous antioxidant enzymes such as superoxide dismutase (SOD), catalase and glutathione (GSH) peroxidase (Px) [23]. The major characteristic of polyphenols is the presence of two or more hydroxyl groups attached to an aromatic ring, which gives them the antioxidant capacity to neutralize the unpaired electrons of free radicals, donate hydrogens or chelate metal ions [24].

Despite the well-known health protective role played by the intake of foods such as fruits and vegetables, especially due to their high polyphenol content [9,10], the population does not achieve the daily requirements recommended by the health authorities, which range between 400 and 800 g/day [25,26]. That fact, together with the growing interest and consumption of natural products based on nutraceuticals [27,28], which in addition to the convenience of consumption have the advantage of lacking side effects compared to drugs [28] was the reason for choosing to use a nutraceutical during the intervention.

In this context, the aim of the present investigation is to evaluate if the long-term consumption of a nutraceutical with high polyphenol content is effective against inflammation and oxidative stress in healthy volunteers of both sexes, by measuring oxylipins in urine by UHPLC-QqQ-ESI-MS/MS, considered the gold standard method.

## 2. Materials and Methods

### 2.1. Chemicals and Reagents

Twelve IsoPs derived from AA (2,3-dinor-15-*epi*-15F_2t_, 2,3-dinor-15-F_2t_-IsoP, 9-*epi*-15-F_2t_-IsoP, 15-*epi*-15F_2t_-IsoP, *ent*-15-*epi*-15F_2t_-IsoP, 15-keto-15-F_2t_-IsoP, 15-F_2t_-IsoP, *ent*-PGF_2α_, 5-F_2t_-IsoP, 5-*epi*-5F_2t_-IsoP, 15-keto-15E_2t_-IsoP, 15-*epi*-15E_2t_-IsoP); two IsoPs derived from DGLA (8-iso-PGF_1α_, 8-iso-PGE_1α_); two IsoPs derived from EPA (8-F_3t_-IsoP, 8-*epi*-8-F_3t_-IsoP); 15 enzymatic metabolites derived from AA (PGD_2_, Tetranor-PGJM, 11β-PGF_2__α_, 2,3-dinor-11β-PGF_2__α_, PGDM, Tetranor-PGDM, PGE_2_, 20-OH-PGE_2_, Tetranor-PGEM, Tetranor-PGAM, 15-keto-PGF_2__α_, Tetranor-PGFM, 20-OH-PGF_2__α_, 19(R)-OH-PGF_2__α_, 11-DH-TXB_2_); two metabolites derived from DGLA (PGE_1_, PGF_1__α_) and one metabolite derived from EPA (17-*trans*-PGF_3__α_) were analysed in the current study. Table 1 shows the different oxylipins categorized according to their metabolization pathway, retention time, transition from precursor ion to product ion and their molecular weight. The authentic standards corresponding to this range of oxylipins were purchased from Cayman Chemicals (Ann Arbor, MI, USA). The enzyme β-glucuronidase, type H2 from *Helix pomatia*, and BIS-TRIS (bis-(2hydroxyethyl)-amino-tris(hydroxymethyl)-methane) were obtained from Sigma-Aldrich (St. Louis, MO, USA). All LC-MS grade solvent were obtained from J.T. Baker (Phillipsburg, NJ, USA). The Strata X-AW, 100 mg per 3 mL solid phase extraction (SPE) cartridges were purchase from Phenomenex (Torrance, CA, USA). Water was treated in a Milli-Q water purification system from Millipore (Bedford, MA, USA).

### 2.2. Clinical Trial Design

This study consisted of a randomized, cross-over, double-blind, sex-stratified, and placebo-controlled clinical trial to assess the effectiveness of daily consumption of an encapsulated nutraceutical (Juice Plus+ Premium^®^, The Juice Plus+ Company, Collierville, TN, USA) based on different berry, fruit, and vegetable blends, on urine inflammation markers (oxylipins). The intervention had a length of 36 weeks, two periods of 16 weeks, separated by a washout period of 4 weeks, as depicted in Figure 1.

During the intervention, the volunteers came to the laboratory four times (at the beginning and the end of each phase), providing the urine samples collected during the last 24 h (from the morning of the previous day until the first urine of the morning of the day of the visit. After delivery of the urine, the researchers noted the 24 h urine volume, shook the contents, and stored a small homogeneous sample in several 2 mL Eppendorf tubes at a temperature of −80 °C, until the analysis was carried out.

At the beginning of each phase, volunteers were provided with the product to be consumed (product under investigation (extract (EXT)) or placebo (PLA)), and at the end of each phase the volunteers had to deliver the remaining product in its original package to check compliance with the process. At visit 0 (V0), the researchers checked the inclusion and exclusion criteria. The selected volunteers signed an informed consent form and consequently a researcher from outside the research group carried out the randomization, using a software application (Epi-dat v4.1 Epi-dat, Galicia, Spain) to assign the volunteers to each of the groups. Neither the researchers nor the volunteers knew which groups the volunteers had been allocated to. Both the EXT and the PLA were labelled with a different code (A and B) and only at the end of the study did the company notify the researchers which code represented the EXT and which one the PLA. The protocol was approved by the Institutional Review committee of the Catholic University San Antonio of Murcia (UCAM) (date: 24 November 2017; code CE111072). This study was carried out following the Standards of Good Clinical Practice and was conducted according to the Declaration of Helsinki. The trial was registered at www.clinicaltrials.gov (accessed on 11 June 2020) (identifier CFE/JU/44-17). The study was carried out in the Pharmacy Department of the Faculty of Health Sciences of the Catholic University San Antonio of Murcia (UCAM). Current European legislation on the protection of personal data was complied with (Regulation (EU)2016/679).

### 2.3. Participants

The volunteers had to meet all the inclusion criteria (body mass index (BMI) between 18.5 and 35 kg/m^2^, consume no more than three servings of fruit and vegetables per day, no chronic disease, age between 18–65 years, and sign the informed consent) and not meet any exclusion criteria (being under medical treatment, allergy to fruits or vegetables, being on a diet, being vegetarian or vegan, smoking, consumption of more than three glasses of alcohol (wine and beer) per day, pregnant, having donated blood (0.5 L) or having undergone a major surgery in the last 3 months, having sleep problems, and taking multivitamins). After recruiting a total of 117 volunteers and checking the inclusion and exclusion criteria, a total of 108 volunteers initiated the study. During intervention, 16 volunteers were lost to follow-up and 92 completed the intervention.

### 2.4. Test Supplement

The EXT and PLA presented similar appearance characteristics and were manufactured and supplied by The Juice Plus+ Company, Collierville, TN, USA. The EXT and PLA were encapsulated in the form of pills, provided in bottles only differentiated by the codes.

The tested EXT was composed of 36 types of berries, fruits and vegetables lyophilized and encapsulated powder, well detailed in the following publication [29]. The daily dose was six capsules/day, divided into three in the morning preferably fasting, and three in the middle of the afternoon with their evening meal, providing 600 mg of polyphenols, 159 mg of vitamin C, 18.7 mg of vitamin E, 6.1 mg of lutein, 2.91 mg of carotene, 1 mg of lycopene, 0.15 mg of astaxanthin and 318 µg of folate. The PLA was composed of microcrystalline cellulose, rice starch, vegetable capsule (cellulose), magnesium stearate and artificial colouring.

Through UHPLC-QqQ-MS Bresciani et al. performed the characterization of the EXT, detecting a total of 119 phenolic compounds belonging to different phenolic families [30]. According to another investigation on bioavailability of the EXT carried out by Bresciani et al., 20 quantifiable metabolites were identified [31].

### 2.5. Study Variables

All the variables were analysed four times during the study: At the beginning and at the end of each 16 week product consumption phase. During the intervention, the volunteers were asked to maintain their habits and were not allowed to begin or modify any hormonal treatment, nor make any significant dietary or physical activity changes that could affect the study variables. In addition, subjects had to come in after a 12 h fasting period, and were only allowed water intake in the 3 h prior to the visit, in order to perform other complementary tests (blood samples and cognitive test) as can be found in the following publications [29,32].

### 2.6. Extraction of Human Oxylipins in Urine Samples

To carry out the extraction of human oxylipins in urine, an enzymatic hydrolysis step simulating digestion was performed beforehand, followed by solid-phase extraction described below [33,34].

Once the urine samples were collected from each subject, they were stored at −80 °C until analysis. For enzymatic hydrolysis, the samples were allowed to thaw, and 1 mL of urine was taken at room temperature to which 100 µL of 0.1M acetate buffer, pH 4.9 was added, then 55 µL of enzyme (β-glucuronidase *Helix pomatia* G-0876) was added in order to eliminate the glucuronide and sulphate conjugates according to [35,36] and incubated for 2 h at 37 °C in warm bath. After two hours, 500 µL of mM MeOH/HCL was added to cause a protein precipitate, vortexed and centrifuged at 10,000 rpm for 5 min.

Subsequently, it was passed to phase two solid phase extraction (SPE). For sample processing, the supernatant of the samples was transferred to 15 mL falcon tubes, 1250 µL of MeOH was added and vortexed. Two mL of Bis Tris 0.02 M HCL, pH 7 buffer was added and vortexed again. The next step was to condition the cartridges (Stra-ta-X-AW- Phenomenex) and then load the sample. To condition the cartridges, 2 mL of MeOH was added and 2 mL of MiliQ H_2_O was extracted, the sample was loaded and washed with 4 mL of MiliQ H_2_O and the air flow removed the non-intersecting compounds from the cartridges. Subsequently, the elution of the compounds of interest was performed with 1 mL of MeOH and stored in an eppendorf to later dry the eluent in the speed vacuum concentrator for 24 h and stored frozen until further analysis.

Prior to UHPLC analysis, the sample had to be reconstituted with 200 µL of mobile phase (H_2_O/0.1% formic acid:MeOH, 90:10), sonic bath for 10 min and filtered through 0.45 µL PTFE filters with 1 mL syringes to finally introduce the sample into amber vials glass with insert.

### 2.7. Analysis UHPLC-QqQ-MS/MS of Oxylipins

Chromatographic separation of oxylipins present in urine was performed employing a UHPLC acylated to a 6460 QqQ-MS/MS (Agilnet Technologies, Waldbronn, Germany), using the configuration described previously [4,33]. Resuming chromatographic separation was carried out on an ACQUITY BECH C_18_ columns (2.1 × 150 mm, pore size of 1.7 μm) (Waters, MA, USA). MS analysis was applied in negative MRM ESI mode. The mobile phases used were solvent A (Milli-Q H_2_O/acetic acid (99.99:0.01, *v*/*v*)) and solvent B (MeOH/acetic acid (99.99:0.01, *v*/*v*)). The flow rate and injection volume were 0.150 mL/min and 20 μL/min, respectively. Identification and quantification of oxylipins was achieved by analysis of the parental masses and oxylipin-specific fragmentation patterns Table 1. Performed by multiple reaction monitoring mode (MRM) mass spectrometry analysis and the application of electrospray ionization. Moreover, identification and quantification of the monitored oxylipins was performed using authentic standards of the oxylipins referenced in Table 1. Data acquisition and processing were performed with MassHunter software version B.08.00 (Agilent Technologies, Waldbronn, Germany).

### 2.8. Statistical Analysis

Statistical analysis was previously described [29]. The results of the variables were presented as mean ± standard deviation (SD) at baseline and at the end of the intervention. Kokmogorov–Smirnov was applied to verify that the normality T-Student was carried out to compare the two branches of the study, repeated measures analysis of variance with time as an intrasubject factor. A Bonferroni test was applied for post-hoc analysis. The SPSS 24 computer software (SPSS, Inc., Chicago, IL, USA) was employed and a significance level of 0.05 was chosen.

## 3. Results

### 3.1. Study Population

As the flow diagram in Figure 2 depicts, a total of 117 volunteers were selected to begin the study, of which 108 began the intervention and were divided into two homogeneous groups. During the intervention and after loss to follow-up and dropouts a total of 92 volunteers completed the treatment. Table 2 shows the demographic data.

### 3.2. Oxylipins

The urinary concentration of oxylipins analysed throughout the intervention is shown in Table 3. At the beginning of the study, no significant differences were observed for any of the variables analysed, showing a well performed stratification of the volunteers. Figure 3 summarizes the different routes followed by the oxylipins quantified in the current study.

#### 3.2.1. Prostanoids

Regarding the prostanoids derived from arachidonic acid for D pathway, data of the three oxylipins quantified showed a similar trend in terms of reduction. From PGD_2_, a decrease was observed after consumption of both PLA Δ (−0.083) and EXT Δ (−0.071) without being statistically significant in any group *p* > 0.05. For 11-*β*-PGF_2_α, a decrease was observed in both groups PLA Δ (−0.048) vs. EXT Δ (−0.215), being only significant in the EXT group (*p* < 0.05). Finally, in 2,3-dinor−11β-PGF_2_α, a slight increase Δ (−0.037) not significant p = 0.692 was observed in PLA and a significant reduction (*p* < 0.05) was observed after EXT Δ (−0.145). After comparison between groups at the end of the study, none of the three oxylipins of D pathway showed significant differences.

Regarding the E pathway, a slight increase in PGE_2_ was observed after consumption of PLA Δ (0.085) without statistical significance, showing a statistically significance increase (*p* < 0.05) after consumption of EXT Δ (0.241). In this case, comparing between groups at the end of the study, statistically significant differences (*p* < 0.05) were observed.

Regarding the F pathway, 15-keto-PGF_2_α presented a reduction after consumption of both PLA Δ (−0.047) and EXT Δ (−0.158). As with PGE_2_ in the PLA it was insignificant being statistically significant after EXT *p* < 0.005 as well as comparing between groups at the end of the study (*p* < 0.05).

Regarding DGLA-derived prostanoids PGF_1__α_, despite varying in different ways with a slight increase after PLA Δ (0.005) and a reduction after consumption EXT Δ (−0.054), no significant statistical differences were found at the final comparison between groups.

#### 3.2.2. Isoprostanes

All IsoPs that we were able to quantify in the present research correspond to the 15 series derived from AA.

The 2,3-dinor-15-*epi*-15F_2t_ decreased in both groups PLA Δ (−0.032) EXT Δ (−0.057) showing statistical significance only after the consumption of EXT (*p* < 0.05). However, the final comparison between groups did not show significant differences between groups.

The evolution of 2,3-dinor-15-F_2t_-Isop was inverse, in PLA it increased Δ (0.068) and in EXT it decreased Δ (−0.048) despite not being significant in either of the two groups. However, due to that it increased in one group and decreased in the other, significant differences were observed comparing both groups at the end of the study (*p* < 0.05).

The 15-*epi*-15F_2t_ in both groups showed a slight increase in PLA values Δ (0.084) vs. EXT (0.052), with no statistical significance neither in the intra- nor in the inter-group evolution at the end of the study (*p* < 0.05).

With respect to PGF*_2_*α and as with 2,3-dinor-15-*epi*-15F_2t_, it varied in the opposite way after consumption of PLA Δ (0.100) vs. EXT Δ (−0.151) despite not being significant in either of the two groups. However, due to that it increased in one group and decreased in the other, significant differences were observed comparing both groups at the end of the study (*p* < 0.05).

Figure 4 depicts the variation observed in the total content of prostanoids, IsoPs and oxylipins. After consumption of PLA, prostanoids remained almost stable with only a slight non-significantly increase Δ (0.13), whereas the total values of IsoPs Δ (0.76) and oxylipins (0.89) increased significantly. On the other hand, after EXT consumption, reductions were observed in all three Δ (−0.34; −0.44; −0.78) for prostanoids, IsoPs and total oxylipins, respectively being significant in the last two (*p* < 0.05). Nevertheless, comparing between groups at the end of the intervention, significance was observed for all three (*p* < 0.05). Additionally, comparing the evolution of the groups during the intervention, it was observed that for both total oxylipin and total IsoPs content, there were statistically significant differences (*p* < 0.001), while for prostanoids content, no such significance was observed (*p* = 0.158).

## 4. Discussion

The major strength of the present research relies on the fact that long-term supplementation with a natural and high polyphenolic nutraceutical decreases total levels of oxylipins, isoprostanes and prostanoids with associated improvements in human health.

The main obstacle that arises in assessing certain polyphenol-based products is the fact that there is a wide range of products on the market that declare certain components in their labelling that they do not contain [37,38]. In the current research, and thanks to previous publications concerning the characterization and bioavailability of the EXT [30,31], it can be affirmed that the EXT used contains a high phenolic content (600 mg of total phenolics comprising ellagitannins, anthocyanins, gallotannins, dihydrochalcones, flavan-3-ols, flavone, flavonols, flavanone, hydroxybenzoic acids, hydroxycinnamic acids, phenylethanoids, lignans, and glucosinolate) [30], even considerably higher than that observed in certain foods [39]. All metabolites quantified in plasma were found in conjugated form with glucuronide, glycine or sulphate and were metabolized and absorbed at different times along the digestive tract, especially in the colonic microbiota [31]. As far as it is known, the antioxidant capacity of foods/supplements is related to the antioxidant capacity of the plasma [40,41].

Compared to the consumption of vitamins, that show short-term health benefits after ingestion, evidence shows that a moderate long-term consumption of polyphenols is required in order to achieve benefits [42]. Despite the known beneficial effects of polyphenols, the scientific literature shows a low bioavailability of polyphenols in humans [43]. However, the combined intake of different polyphenols, as is the case of the EXT under investigation, seems to have greater efficacy compared to the intake of isolated compounds [44].

Phenolic compounds are characterized by the presence of at least one aromatic ring with one or more hydroxyl groups attached and are classified as flavonoids and non-flavonoids [20,45,46]. Of the major phenolic compounds present in the EXT, five flavonoids (flavan-3-ols, flavonols, flavones, flavanones and anthocyanins) and one non-flavonoid (ellagitannins) stand out [31]. Most of the protective effects of flavonoids in biological systems are attributed to their ability to transfer electrons to free radicals, chelate metal catalysts, activate antioxidant enzymes, reduce alpha-tocopherol radicals, and inhibit oxidases [47]. In humans, flavonoids (main compounds of the EXT), are known to have a broad spectrum of health benefits, exhibiting anticancer, cardioprotective, anti-aging, anti-inflammatory, neuroprotective, antidiabetic, immunomodulatory, antiviral, antiparasitic, and antimycobacterial properties [48,49,50].

Through the inhibition of the synthesis of certain proinflammatory oxylipins or the overexpression of other anti-inflammatory oxylipins, inflammation can be modulated. Polyphenols are effective in inhibiting certain enzymes involved in the synthesis of proinflammatory oxylipins such as PLA2, COX or LOX. It should be considered, that this type of compound is really difficult to evaluate given their low concentration range that oscillates from picograms to nanograms in most biological samples [51].

The choice of measuring IsoPs to assess oxidative damage is because they are a reliable biomarker of endogenous lipid peroxidation, and they are chemically stable in biological fluids. Although they can be measured in different fluids, the fluids of choice are usually plasma and urine because their collection is less invasive, although precautions should be considered for the storing of samples [52]. As regards the method of choice, urinary concentration of IsoPs is considered the “gold standard” method for assessing the redox status of the organism [36]. The F_2_IsoPs are metabolized and excreted in urine relatively quickly, and their quantification is of real interest and useful for monitoring the oxidative state over a specific period of time [33]. As their free form represents only a part of the total F_2_IsoPs, it is necessary to carry out a hydrolysis prior to their analysis in order to release the esterified compounds. Because they are excreted as glucuronides, prior processing with β-glucuronidase considerably increases their urinary levels [53].

The molecules IsoPs are derived from AA, they are similar to prostaglandins, but unlike prostaglandins, they are synthesized by a COX-independent pathway through a non-enzymatic ROS-dependent peroxidation, forming a compound similar to PGG_2_ [52,54]. In their synthesis and metabolization, IsoPs originate from lipid AA as opposed to COX-derived PGs that make free AA [55].

The F_2_IsoP markers are considered to be the most reliable for monitoring oxidative stress [56]. It is well known that elevated concentrations of these are associated with increased risk of cardiovascular disease as well as worse prognosis [57] and that they are elevated in multiple pathological conditions in humans, being highly elevated in obese volunteers, a pathology associated with chronic low-grade inflammation, in recently operated patients, and in cancer patients [52]. Therefore, any treatment capable of reducing the total content of IsoPs could contribute to improve health. According to the results section of the present investigation, at the end of the intervention, total IsoPs, total prostanoids and, therefore, total oxylipins are significantly reduced after the consumption of EXT versus PLA, while both IsoPs and total oxylipins are increased after the consumption of PLA. These results, together with the fact that when comparing the evolution between groups at the end of the intervention, significant differences were observed for IsoPs and total oxylipins, seem to imply that EXT intake can reduce oxidative stress in the organism.

Regarding IsoPs, only those belonging to the 15 series could be quantified. The 15-F_2_Isop (the primary IsoP) could not be quantified, which may be due to its rapid metabolization. Nevertheless, four metabolites of its precursor 15-F_2t_Isop could be quantified (Table 3). The results showed a significant reduction in three of them, with the exception of 15-*epi*-15-F_2t_Isop, not showing significance. Its precursor, 15-F_2_Isop, is known as potent vasoconstrictor in most vascular beds, inhibits angiogenesis, modulates platelet activity, and induces atherosclerosis by stimulating neutrophil and monocyte adhesion in endothelial cells [52]. According to the results where the total IsoPs are significantly reduced, and that all of them are metabolites of 15-F_2_Isop, we could foresee that the intake of the polyphenolic combination produces improvements at the cardiovascular level.

In an intervention on the consumption of wine and grape must [36], a reduction in the total content of IsoPs was also observed, being more pronounced in wines than in musts, due to the bioactive compounds present in these products.

Prostanoids:

The PGs are isoprostanoids whose biological functions have been widely documented, and according to their pathway/family, different actions are attributed to them in the organism, and exert their effects through activation of G-protein-coupled receptors [6]. The PGs are synthesized from AA by the action of the COX enzyme, forming PGG_2_, then PGH_2_ and from this, depending on the route (E, I, D, F) are synthesized the primary PGs; PGE_2_, PGI_2_, PGD_2_, PGF_2α_ [4].

Pathway E:

In the current research, the values of PGE_2_ increased after the EXT consumption versus PLA. In inflammatory processes and the classic processes of pain, redness and swelling, it has been seen that it can present both pro-inflammatory or anti-inflammatory, pro/anti-thrombotic effects depending on the E-prostanoid (EP) receptors (EP_1_, EP_2_, EP_3_, EP_4_) [5]. In early stages of inflammation, PGE_2_ tends to be proinflammatory, whereas in subsequent resolution of inflammation it tends to be anti-inflammatory [58]. The PGE_2_ is able to attenuate the production of TNF-α and suppress the production of pro-inflammatory cytokines such as IL-6 [59]. In the current research, the increase in its value after EXT consumption together with the observed reduction in total prostanoids content seems to indicate that it plays an anti-inflammatory role.

Pathway D:

Despite not observing significant differences in the primary PG (PGD_2_), significant decreases in its metabolites (11-*β*-PGF_2α_ and 2,3-dinor-11β-PGF_2α_) were observed after EXT consumption. The compound 2,3-dinor-11β-PGF_2α_ is itself a metabolite of 11-β-PGF_2α_ [60] and both exhibit similar biological activity that inhibits adipose tissue differentiation and increases bronchoconstriction [36]. Similar to PGE_2_, PGD_2_ can have both anti-inflammatory and pro-inflammatory capacities [61]. Considering that PGD_2_, moreover to being synthesized through PGH_2_, could be synthetized through PGE_2_ [52], it can be deduced that if there is a decrease in the urinary excretion of PGD_2_ metabolites (11-β-PGF_2α_ and 2,3-dinor-11β-PGF_2α_) with a proinflammatory character, together with an increase in PGE_2_, without affecting the total concentration of PGD_2_, the increase in the excretion of PGE_2_ observed in the previous section is of an anti-inflammatory character.

Pathway F:

In this pathway, only one PGF_2α_ metabolite, 15-keto-PGF_2α_, could be quantified, which showed a significant reduction after consumption of polyphenolic EXT not observed after consumption of PLA. According to the scientific literature, PGF_2α_, is associated with proinflammatory states by interfering in the synthesis and secretion of TNF-a and proinflammatory cytokines such as IL-1β, IL-6, IL-8, with a vasoconstrictor effect [62]. Therefore, the significant decrease of 15-keto-PGF_2α_ (its only quantified metabolite) suggests a strong anti-inflammatory potential.

In an intervention consisting of the intake of half or full broccoli ration versus control [63], no significant variations in total IsoPs excretion content were observed, but a decrease in prostanoids excretion, like 11-β-PGF_2α_, was. Such contradictory data could be explained due to both the small sample size (*n* = 24) and the short duration of the intervention (3 days), suggesting long-term consumption protocols with a considerable sample size—as the present investigation—as the most suitable protocols for those type of studies. The strengths of the present research are the sample size, which is larger than that observed in similar research, which gives us greater statistical power, as well as the duration (four months) of the intervention.

In a study on the total concentration of IsoPs, lasting 6 weeks, no significant reductions were observed after the intake of grapefruit vs. PLA. They were only observed after analysing a subsample of the obese population in the group that consumed grapefruit [64].

Another study on oxylipins and polyphenols (Aronia juice) in triathletes [4], showed reductions of 11-β-PGF_2α_ after consumption of Aronia juice, and on the other hand, increases in other metabolites such as PGE_2_ or 15-keto-15F_2t_-IsoP. However, it does not seem appropriate to compare the data with the present study given that the sample chosen (athletes) is not similar to that of the present study, and many of the effects observed are due to the practice of physical activity itself. In addition, the duration of their intervention was also considerably shorter (45 days) than that of the present study, and the product was not similar to ours.

The present results observed in urine concur with those observed in previous publications on plasma parameters of oxidative stress, where reductions in oxidized LDL, CRP, TNF-1, homocysteine, TNF- and increased HDL-cholesterol were observed after consumption of the product, and they could be related to the improvements observed in cognitive function [29,32].

On the other hand, one of the limitations of the intervention may have been the choice of a healthy population that does not present a high level of certain IsoPs, which is why it may not have been possible to quantify them at baseline; for example, if we had chosen an obese population or a population with any of the pathologies described in the manuscript, much more pronounced changes would have been observed.

## 5. Conclusions

The current research data suggest that the long-term consumption, for 4 months, of an extract rich in polyphenols based on a combination of berries, fruits, and vegetables, is effective in dealing with oxidative stress, in a population without previous pathologies. The reduction of oxidative stress seems to be produced by the attenuation of oxylipins derived from arachidonic acid such as PGs of the D and F family, known to present a high inflammatory component, and increasing PGE_2_, which can behave both as proinflammatory and anti-inflammatory. Furthermore, the significant reduction of total F_2_IsoPs seems to reinforce our hypothesis, specifically the variation observed in 15 series whose precursor and primary Isoprostane is 15-F_2_Isop, characterized by pro-inflammatory and vasoconstrictor capacity.

## Figures and Tables

**Figure 1 antioxidants-11-01342-f001:**
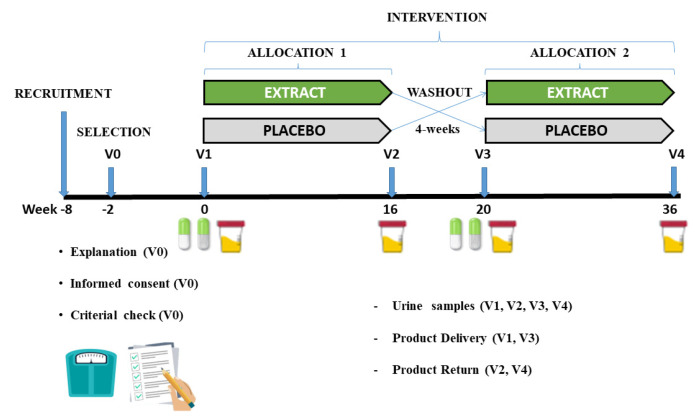
Graphical representation of the different phases of the study.

**Figure 2 antioxidants-11-01342-f002:**
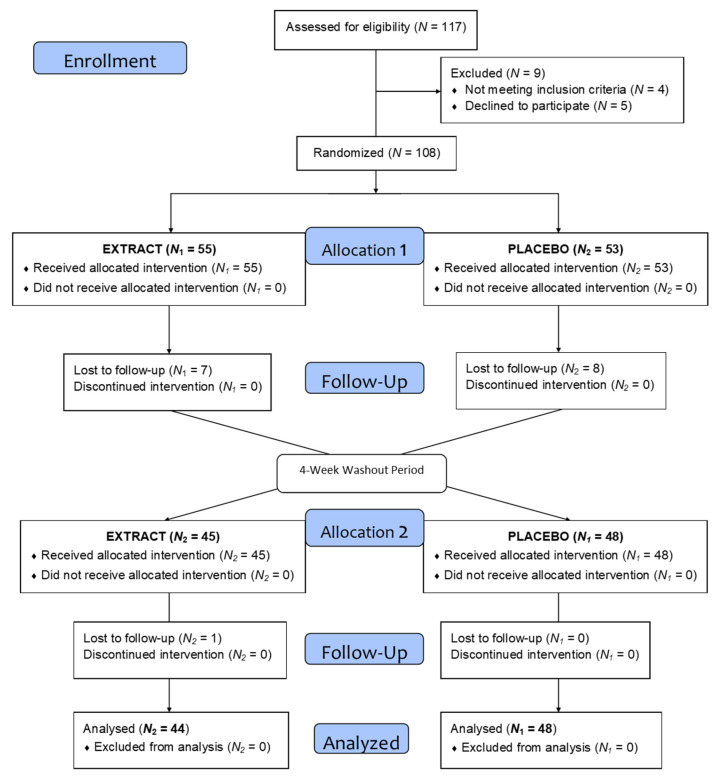
Flow diagram.

**Figure 3 antioxidants-11-01342-f003:**
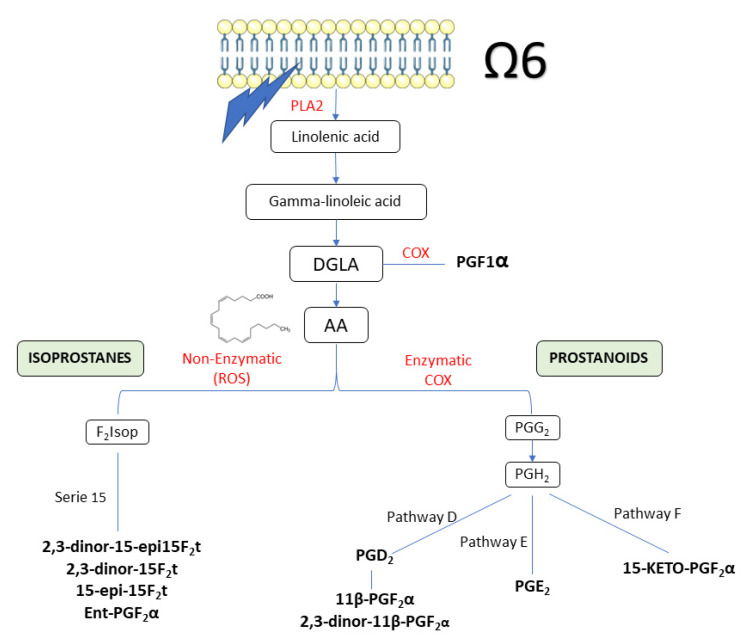
Summary of the pathway of oxylipins quantified in the current study. Oxylipins are in bold. AA: arachidonic acid; DGLA: dihomo-γ-linoleic acid; PLA2: Phospholipase 2.

**Figure 4 antioxidants-11-01342-f004:**
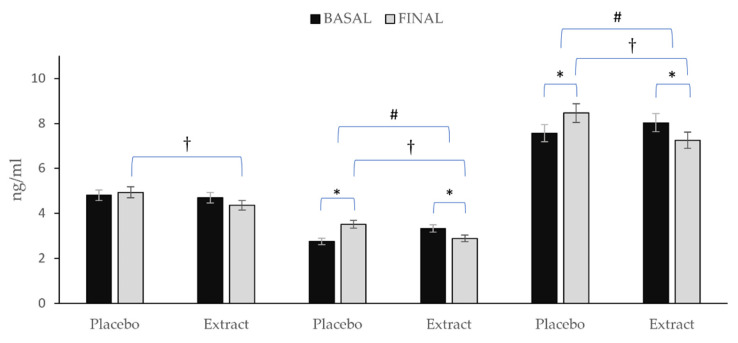
Variation in total prostanoids, total IsoPs, and total oxylipins in each group during the intervention. * Means significant statistical differences comparing the evolution between baseline and final intragroup (*p* < 0.05). † Means significant statistical differences comparing final moment between groups at the end of the intervention (*p* < 0.001). # Means significant statistical differences comparing evolution between groups during the intervention (*p* < 0.05).

**Table 1 antioxidants-11-01342-t001:** Oxylipins analysed in the current assay.

Oxylipins	Retention Time (min)	MRM Transition (*m*/*z*)	MW (g/mol)
**Isoprostanes**			
**Isoprotanes generated from AA**			
**F_2_IsoP (15-Serie)**			
2,3-dinor-15-*epi*-15F_2t_ (*)	8.29	324.9 > 237.1	326.43
2,3-dinor-15-F_2t_-IsoP (*)	8.84	325.2 > 237.1	326.4
9-*epi*-15-F_2t_-IsoP	11.75	353.2 > 193.1	354.5
15-*epi*-15F_2t_-IsoP (*)	11.77	353 > 193	354.5
*ent*-15-*epi*-15F_2t_-IsoP	11.77	353.1 > 309.1	354.5
15-keto-15-F_2t_-IsoP	11.8	351.1 > 315.1	352.5
15-F_2t_-IsoP	12.3	353.2 > 193.1	354.5
*ent*-PGF_2α_ (*)	13.6	353.1 > 309	354.5
**F_2_soP (5-Serie)**			
5-F_2t_-IsoP	12.82	353.1 > 335.2	354.48
5-*epi*-5F_2t_-IsoP	13.53	353.2 > 334.8	354.48
**E_2_IsoP (15-Serie)**			
15-keto-15E_2t_-IsoP	12.04	349 > 234.9	350.5
15-*epi*-15E_2t_-IsoP	12.94	355.1 > 315.1	352.47
**Isoprotanes generated from EPA**			
8-F_3t_-IsoP	10.55	351.2 > 126.8	352.47
8-*epi*-8-F_3t_-IsoP	11.23	350.6 > 127.2	352.47
**Isoprotanes generated from DGLA**			
8-iso-PGF_1α_	12.14	355.1 > 311.1	356.5
8-iso-PGE_1α_	12.85	352.9 > 234.8	354.5
Prostanoids			
**Prostaglandins generated from AA**			
**Prostaglandins D-pathway**			
Tetranor-PGDM	3.17	327.1 > 108.9	328.4
PGDM	3.2	327.1 > 309.1	328.4
Tetranor-PGJM	3.6	309 > 155	310.3
2,3-dinor-11β-PGF_2__α_ (*)	10.57	325.2 > 237.1	335.5
PGD_2_ (*)	13.22	350.9 > 315	352.5
11β-PGF_2__α_ (*)	13.61	353 > 309.1	354.5
**Prostaglandins E-pathway**			
Tetranor-PGEM	3.17	327 > 308.8	328.4
Tetranor-PGAM	3.58	309 > 290.9	310.3
20-OH-PGE_2_	4.66	367.1 > 349.2	368.5
PGE_2_ (*)	13.07	351.2 > 333.1	352.5
**Prostaglandins F-pathway**			
Tetranor-PGFM	3.14	329.4 > 311.1	330.4
20-OH-PGF_2__α_	5.1	369.2 > 325.1	370.5
19(R)-OH-PGF_2__α_	5.15	369.2 > 325.1	370.5
15-keto-PGF_2__α_ (*)	12.9	351.1 > 314.9	352.5
**Thromboxane**			
11-DH-TXB_2_	12.32	367 > 161.1	372.5
**Prostaglandins generated from EPA**			
17-*trans*-PGF_3__α_	12.47	350.9 > 307	352.5
**Prostaglandins generated from DGLA**			
PGE_1_	12.84	353.2 > 317.2	354.5
PGF_1__α_ (*)	13.67	355.2 > 311	356.5

AA: arachidonic acid; EPA: eicosapentanoic acid; DGLA: dihomo-γ-linolenic acid; MRM: multiple reaction monitoring; (*) These oxylipins were quantified in the current assay.

**Table 2 antioxidants-11-01342-t002:** Demographic data.

Variable	Total	N_1_	N_2_
N	92	48	44
Men	45	20	25
Women	47	28	19
Age (years)	34 ± 11	33 ± 10	36 ± 12
Weight (kg)	73.1 ± 14.3	70.7 ± 13.9	75.7 ± 14.4
Height (m)	1.72 ± 9	1.71 ± 9	1.73 ± 9
BMI (kg/m^2^)	24.4 ± 3.4	23.9 ± 3.4	25.0 ± 3.4

**Table 3 antioxidants-11-01342-t003:** Evolution of the different isoprostanes measured during the study. Values are expressed as mean and standard deviation at the beginning and end of the intervention.

Oxylipins (ng/mL)	Product	Baseline	Final	*p-*Value Intragroup	*p-*Value Final
**Prostanoids generated from AA**					
**D pathway**					
PGD_2_	Placebo	0.85 ± 0.21	0.77 ± 0.16	0.065	0.243
	Extract	0.80 ± 0.19	0.73 ± 0.17	0.081	
11-*β*-PGF_2α_	Placebo	1.18 ± 0.38	1.13 ± 0.32	0.704	0.118
	Extract	1.14 ± 0.43	0.92 ± 0.27	0.037 *	
2,3-*dinor*-11β-PGF_2α_	Placebo	1.16 ± 0.31	1.20 ± 0.41	0.692	0.109
	Extract	1.17 ± 0.35	1.02 ± 0.27	0.033 *	
**E pathway**					
PGE_2_	Placebo	1.01 ± 0.30	1.10 ± 0.39	0.342	0.017 ^†^
	Extract	1.02 ± 0.38	1.26 ± 0.38	0.020 *	
**F pathway**					
15-keto-PGF_2α_	Placebo	0.92 ± 0.23	0.88 ± 0.24	0.438	0.045 ^†^
	Extract	0.90 ± 0.25	0.74 ± 0.19	0.004 *	
**Prostanoids generated from DGLA**					
PGF_1α_	Placebo	0.63 ± 0.14	0.64 ± 0.10	0.886	0.182
	Extract	0.64 ± 0.10	0.59 ± 0.15	0.101	
**Isoprostanes generated from AA**					
2,3-dinor-15-*epi*-15F_2t_	Placebo	0.56 ± 0.16	0.53 ± 0.17	0.436	0.680
	Extract	0.60 ± 0.10	0.55 ± 0.08	0.025 *	
2,3-dinor-15-F_2t_-Isop	Placebo	0.60 ± 0.15	0.67 ± 0.10	0.098	0.002 ^†^
	Extract	0.61 ± 0.12	0.56 ± 0.11	0.258	
15-*epi*-15F_2t_	Placebo	1.20 ± 0.31	1.28 ± 0.38	0.500	0.232
	Extract	1.07 ± 0.36	1.12 ± 0.32	0.643	
*ent*-PGF_2α_	Placebo	1.62 ± 0.39	1.72 ± 0.51	0.513	0.017 ^†^
	Extract	1.59 ± 0.37	1.44 ± 0.32	0.137	
**Total**					
Total Prostanoids	Placebo	4.81 ± 1.43	4.94 ± 1.50	0.596	0.017 ^†^
	Extract	4.70 ± 1.33	4.36 ± 0.91	0.105	
Total IsoPs	Placebo	2.76 ± 1.24	3.52 ± 1.08	<0.001 *	<0.001 ^†^
	Extract	3.33 ± 1.24	2.89 ± 0.96	0.025 *	
Total Oxylipins	Placebo	7.57 ± 2.11	8.46 ± 2.07	0.021 *	<0.001 ^†^
	Extract	8.03 ± 1.86	7.25 ± 1.23	0.006 *	

* Means significant statistical differences comparing the evolution between baseline and final intragroup (*p* < 0.05). ^†^ Means significant statistical differences comparing the final moment between groups at the end of the intervention (*p* < 0.05). All units are in ng/mL.

## Data Availability

The data are contained within the article.

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
