# Peer review of "Anti-Inflammatory and Antioxidant Capacity of a Fruit and Vegetable-Based Nutraceutical Measured by Urinary Oxylipin Concentration in a Healthy Population: A Randomized, Double-Blind, Placebo-Controlled Clinical Trial"

_antioxidants, 2022, doi:10.3390/antiox11071342_

Round 1

Reviewer 1 Report

General Comment

 This and interesting report regarding the long-term effect of anti-inflammatory and antioxidant capacity of a fruit and vegetable-based nutraceutical in healthy individuals. The authors are providing  an informative and well written introduction, which is not true for section Results. Additionally, some details and already published results from the same study have not been included.

Specific points

1) The test supplement and the placebo should be described in more detail (contents, regimen, etc.)

2) The text of section Results is mainly repeating the content of Table 3. Authors should concentrate on important data (significant changes) within the text.

3) Discussion: the possible effects other compounds other than polyphenols should be considered within the Discussion.

4) The same study included various other parameters as blood parameters and cognitive function already published (references 30 and 33). The main effects (especially plasma parameters of oxidative stress) should be mentioned within this manuscript and compared to results from urinary analyses, which would considerable increase the significance of presented results.

5) Parentheses in Figure 4 regarding the comparison of final results between extract and placebo should be positioned at the respective grey columns.

6) Some significant alterations in oxylipins have been observed within the placebo group (Table 3); this phenomenon has to be considered for evaluation of the nutraceutical effect on respective parameters and should also be mentioned within the discussion.

Reviewer 2 Report

The manuscript presents the results of a study conducted on the effect of a polyphenolic extract administered as a nutraceutical on the occurrence of inflammation in the human body. This effect was determined by measuring the values of selected oxylipins found in urine. The manuscript is interesting, but there are some parts that need improvement.

Major remarks

Table 1 presents information about 16 isoprostanes and 18 prostanoids, with the indication that only selected ones (specifically 10) were labeled in this manuscript. So what was the purpose in presenting them all?

Section 2.4 - Information on what the placebo composition was is missing.

Figure 1 - the information presented here shows that half of the volunteers who took part in the study in the second round received the extract, while the other half received a placebo. Figure 2 - in the second round both groups received placebo. This information is contradictory.

Moreover, it seems that the group taking the extract in the first round received placebo in the second round, while the group taking placebo in the first round received the extract in the second round. This way of conducting the experiment would suggest that giving the placebo also affects the level of inflammation, but I don't think that was the point. Please clarify.

It would be better if the data in Table 3 were ordered as described in Sections 3.2.1 and 3.2.2.

Minor remarks

Line 45 - what does AA stand for?

Line 101 - the part that starts with "Error!" should be removed.

Line 102 - the beginning of the sentence is missing

Table 1 – instead of "isoprotanes generated from AA, EPA, GGLA" should be "isoprostanes generated from AA, EPA, GGLA"

Line 335 - instead of "falvan-3-ols, favonols" should be "flavan-3-ols, flavonols"

Round 2

Reviewer 2 Report

I thank the Authors for answering my questions fully and making the corrections I suggested. I have no further questions.

Regarding the two comments from my previous review, about which the authors had doubts, I already explain what I meant. In the PDF file, on line 102, I see the text "Error! Reference source not found." However, in line 103 there is a sentence without a beginning. It is possible, in the WORD version this is not visible.

As for Table 1, I was referring to the missing letter "s" in the word "isoprotanes".

Author Response

We are grateful for your comments, regarding line 102 we have made a small modification to make it appear correctly in the pdf, and regarding table 1 we have reviewed both the pdf and the word version and we believe that it is corrected.